# Clarithromycin-Loaded Poly (Lactic-*co*-glycolic Acid) (PLGA) Nanoparticles for Oral Administration: Effect of Polymer Molecular Weight and Surface Modification with Chitosan on Formulation, Nanoparticle Characterization and Antibacterial Effects

**DOI:** 10.3390/polym11101632

**Published:** 2019-10-09

**Authors:** A. Alper Öztürk, Evrim Yenilmez, Mustafa Güçlü Özarda

**Affiliations:** 1Department of Pharmaceutical Technology, Faculty of Pharmacy, Anadolu University, 26470 Eskişehir, Turkey; evrimakyil@anadolu.edu.tr; 2Department of Pharmaceutical Microbiology, Faculty of Pharmacy, Anadolu University, 26470 Eskişehir, Turkey; mgozarda@anadolu.edu.tr

**Keywords:** Clarithromycin, PLGA, chitosan, nanoparticles, molecular weight, surface modification, antibacterial activity

## Abstract

Clarithromycin (CLR) is a member of the macrolide antibiotic group. CLR has low systemic oral bioavailability and is a drug of class II of the Biopharmaceutical Classification System. In many studies, using nanoparticles (NPs) as a drug delivery system has been shown to increase the effectiveness and bioavailability of active drug substances. This study describes the development and evaluation of poly (lactic-*co*-glycolic acid) (PLGA) NPs and chitosan (CS)-coated PLGA NPs for oral delivery of CLR. NPs were obtained by nanoprecipitation technique and characterized in detail, and the effect of three molecular weights (M_w1_: 7.000–17.000, M_w2_: 38.000–54.000, M_w3_: 50.000–190.000) of PLGA and CS coating on particle size (PS), zeta potential (ZP), entrapment efficiency (EE%), and release properties etc. were elucidated. Gastrointestinal stability and cryoprotectant effect tests were performed on the NPs. The PS of the prepared NPs were in the range of 178 to 578 nm and they were affected by the M_w_ and CS coating. In surface-modified formulations with CS, the ZP of the NPs increased significantly to positive values. EE% varied from 62% to 85%, depending upon the M_w_ and CS coating. *In vitro* release studies of CLR-loaded NPs showed an extended release up to 144 h. Peppas–Sahlin and Weibull kinetic model was found to fit best for CLR release from NPs. By the broth microdilution test method, the antibacterial activity of the formulations was determined on *Staphylococcus aureus* (ATCC 25923), *Listeria monocytogenes* (ATCC 1911), and *Klebsiella pneumoniae* (ATCC 700603). The structures of the formulations were clarified by thermal (DSC), FT-IR, and ^1^H-NMR analysis. The results showed that PS, ZP, EE%, and dissolution rates of NPs were directly related to the M_w_ of PLGA and CS coating.

## 1. Introduction

Oral administration remains the most appropriate, useful, and convenient route for the delivery of most pharmaceutical active agents. However, the main problem of many orally administered drugs and drug delivery systems is overcoming several obstacles before reaching their target sites. Nowadays, the research for approaches to improve the oral bioavailability of low permeable and low soluble compounds for oral administration continues. Using nanoparticles (NPs) as drug delivery system has taken place in the literature as one of these strategies [1]. By applying nanotechnology to medicine, NPs have been created to mimic or alter biological process [2]. NPs vary in size, although they are considered to be between 10 and 1000 nanometers (nm) and are generally considered to be between 100 and 500 nm in medical applications [2,3]. By means of size, surface properties, and manipulation of the material used, NPs can be developed into therapeutic intelligent systems with increased bioavailability [2]. NP systems offer many advantages, including: improving the stability of hydrophobic drugs, rendering them suitable for oral administration with increased bioavailability; improving biodistribution and pharmacokinetics, resulting in improved efficacy; reducing side effects as a consequence of favored accumulation at target sites and decreasing toxicity by using biocompatible polymers [4]. In addition, NP systems can deliver the drug to specific tissues and provide controlled release of the drug. So, this targeted and sustained drug delivery decreases the toxicity and increases the patient’s compliance with less frequent dosing [2].

Poly (lactic-*co*-glycolic-acid) (PLGA) is a copolymer widely used as a matrix for NPs due to its biocompatible and biodegradable characteristics. PLGA is obtained by different ratios of lactic acid and glycolic acid during polymerization resulting in different molecular weights (M_w_) and different physical, chemical, and physicochemical properties [5]. The mechanical strength of PLGA is particularly affected by its M_w_. The M_w_ of PLGA directly affects NP properties such as particle size, entrapment efficiency, release properties, and bioavailability [6]. When the studies have examined this, it has been reported that the PLGA NPs prepared with Biopharmaceutical Classification System (BCS) class-II (carvedilol [7], rifampicin [8], clotrimazole [9]), BCS class-III (Doxorubicin [10], Lamivudine [11]) and BCS Class-IV (docetaxel [12], paclitaxel [13]) drugs have increased bioavailability and also extended release.

Chitosan (CS) has been widely used in pharmaceutical and medical areas because of its favorable biological properties such as safety, biocompatibility, biodegradability, low-toxicity, bacteriostatic, fungistatic, hemostatic, anticholesterolemic, and anticancerogenic properties [14]. CS’s mucoadhesive property, when positively charged, allows the interaction with negatively charged membranes and mucosa, promoting a greater interaction, adhesion, and retention of the pharmaceutical form containing CS close to the intestinal epithelium. Also, CS has the ability to temporarily open the tight junctions of the intestinal epithelium, thereby increasing the drug permeability [15].

Clarithromycin (CLR) is a semi-synthetic macrolide antibiotic used in many infectious conditions like upper and lower respiratory tracts infections, skin, ear, and other soft tissues infections caused by different bacterial groups. CLR’s chemical formula is C_38_H_69_NO_13_ and the chemical structure of CLR is presented in Figure 1. CLR is acid stable and has a short half-life (3–4 h) compatible with a twice-a-day administration [16,17]. CLR has low systemic oral bioavailability and is a drug belonging to BCS class II. Therefore, short half-life and poor systemic bioavailability of CLR limit the therapeutic efficacy of CLR in intracellular infections. So, a higher CLR dose applied for a longer time to achieve a therapeutic effect that may lead to side and toxic effects such as hepatotoxicity [16,17].

So far, CLR-loaded PLGA NPs have been the subject of research in some studies. In one of these studies, NPs were prepared with three different drug: polymer ratios using only Resomer^®^ RG 502 (lactic acid: glycolic acid ratio 50:50, average M_w_: 12.000) by nanoprecipitation technique [18]. The same study was then supported by intestinal permeability tests [1]. In another study, CLR-loaded PLGA microspheres were prepared using only one type PLGA (lactic acid: glycolic acid ratio 75:25, M_w_ 15.000–30.000) by modified O/W single emulsion-solvent evaporation technique [19]. In another study, CLR-loaded PLGA NPs were prepared by solvent evaporation technique with only one type of PLGA (lactic acid: glycolic acid ratio 50:50) and different drug: polymer ratios [20]. As can be seen from the previous studies, no studies have examined the effect of M_w_ of PLGA polymer on CLR-loaded NPs. [10]. The lactide:glycolide ratio of the PLGA types used in our study was 50:50. The PLGA polymer types were all different in their physical and appearance properties and the lactide:glycolide ratio can completely change the properties expected from the formulation. It is also the fact that the device and surfactant used in preparing the formulation changes the properties of the prepared formulation [11]. In addition, as can be seen from previous studies, no method used in this study is similar to others and in our study chitosan modification was also made to the formulations.

In this study, we aimed to prepare CLR-loaded PLGA NPs with enhanced bioavailability and increased antibacterial effect with low dose use. For this purpose, we examined three different M_w_ of PLGA and CS coatings. We prepared three different PLGA NPs and three different CS-coated PLGA NPs by the ‘nanoprecipitation’ technique. The effects of the M_w_ of PLGA and CS coating on the properties of NPs such as particle size, zeta potential, entrapment efficiency and release rates are discussed in detail. Once these effects were determined, the release kinetics of both burst effect and total release time were examined based on the release results. The crystalline properties of all NPs were then examined by thermal (DSC), FT-IR, and ^1^H-NMR analysis and finally, antibacterial activity against *Staphylococcus aureus* (ATCC 25923), *Enterococcus faecalis* (ATCC 29212), *Listeria monocytogenes* (ATCC 1911), and *Klebsiella pneumoniae* (ATCC 700603) was determined by microdilution method. Within the scope of the study, using different M_w_ of PLGA and coating the formulations with CS is an innovative approach for CLR-loaded PLGA NPs and will add new information to the scientific world.

## 2. Materials and Methods

### 2.1. Materials

Clarithromycin was a kind gift from Sanovel (Istanbul, Turkey). Resomer^®^ RG 502 H [Poly (d,l-lactide-*co*-glycolide), acid-terminated, lactide:glycolide 50:50, M_w_: 7.000–17.000], Resomer^®^ RG 503 H [Poly (d,l-lactide-*co*-glycolide), acid-terminated, lactide:glycolide 50:50, M_w_: 24.000–38.000], Resomer^®^ RG 504 H [Poly(d,l-lactide-*co*-glycolide), acid-terminated, lactide:glycolide 50:50, M_w_: 38.000–54.000], and Span^®^ 60 were purchased from Sigma-Aldrich (St. Louis, MO, USA). Low M_w_ chitosan [Deacetylated chitin/Poly (d-glucosamine), M_w_: 50.000–190.000 Da, viscosity: 20–300 cP] was purchased from Sigma (Steinheim, Germany). Pluronic^®^ F-68 was purchased from Alfa-Aesar (Kandel, Germany). All other chemicals used were of analytical grade.

### 2.2. Preparation of PLGA Nanoparticles and Surface Modification with Chitosan

PLGA-based NPs were prepared by following the nanoprecipitation technique with some modifications [7]. Briefly, a weighed amount of PLGA (90 mg) was dissolved in 3 mL acetone together with Span^®^ 60 (30 mg). Next, 3 mL of this solution was added dropwise at a rate of 5 mL.h^−1^ into 10 mL of Pluronic^®^ F-68 aqueous solution (0.5%, *w*/*v*) under magnetic stirring. Acetone was then allowed to evaporate at room temperature under magnetic stirring for 4 h. The resulting aqueous dispersion was centrifuged to collect the NPs (11.000 rpm, 45 min, 4 °C) (Hettich Rotina-420R, Tuttlingen, Germany). After the NPs were collected, 5 mL of distilled water was added in order to wash the particles. The NPs dispersed in water were again subjected to the above-mentioned centrifugation process. This process was repeated twice to wash the NPs.

For CLR-loaded PLGA-based NP preparation, briefly, the procedure started by adding 9 mg CLR to organic phase. Then, 3 mL of such solution with drugs were added drop-wise at rate 5 mL.h^−1^ into 10 mL of Pluronic^®^ F-68 aqueous solution (0.5%, *w*/*v*) under magnetic stirring. Acetone was then allowed to evaporate at room temperature under magnetic stirring for 4 h. The resulting aqueous dispersion was centrifuged to collect the NPs (11.000 rpm, 45 min, 4 °C) (Hettich Rotina-420R, Tuttlingen, Germany). This process was repeated twice to wash the NPs.

The above procedure was applied with minor modifications when preparing CS-coated formulations. In the CS-coated formulations, the aqueous phase consisted of 10 mL of CS solution (0.25%, *w*/*v*) and Pluronic^®^ F-68 (0.5%, *w*/*v*), both prepared in 2% acetic acid (*v*/*v*) [15,21,22]. All remaining procedures are the same as above and the formulation ingredients are presented in Table 1.

### 2.3. Characterization of Nanoparticles

#### 2.3.1. Particle Size, Polydispersity Index, Zeta Potential

The particle size (PS) and polydispersity index (PDI) of NPs were measured using dynamic light scattering technique on the Zetasizer Nano (Zetasizer Nano ZS, Malvern Instruments, Malvern, UK). PS and PDI of NPs prepared were measured by dispersing the formulation in distilled water. Zeta potential (ZP) was determined using the same instrument in a disposable folded capillary zeta cell at 25 °C room temperature and diluted with distilled water. For statistical analysis all samples were measured in triplicate and the average values and standard deviation of the measurements were calculated.

#### 2.3.2. Assessment of Cryoprotectant Effect on Nanoparticles

Some experiments were carried out on PLGA NPs and CS-coated PLGA NP formulations to determine storage and lyophilization conditions. Following PS measurement of fresh formulations, they were centrifuged and the supernatants were discarded. The resulting particles were added to 1 mL of 5% (*w*/*v*) trehalose solution and the PS measurement was performed again. The dispersion was then divided into five equal portions (200 μL NP suspension) in 5 Eppendorf tubes. While no trehalose solution was added to tube 1, 150 μL, 300 μL, 450 μL, and 600 μL trehalose solution (5% *w*/*v*) were added to other tubes, respectively. All formulations were then frozen at −20 °C and then lyophilization (Scanvac CoolSafePro Labogene, Lillerød, Denmark) was performed for all tubes, the dry particles were removed from the machine and dispersed in 1 mL of water, followed by PS analysis [5,7].

#### 2.3.3. Evaluation of Gastrointestinal Stability of Nanoparticles

Before testing the gastrointestinal stability of NPs, solutions simulating gastrointestinal fluids were prepared. The solutions were pH 1.2 solution; intestinal fluid phosphate buffer solution (pH 6.8); phosphate buffer solution (pH 7.4); and distilled water. All three solutions and distilled water were placed in a shaking water bath at a stirring speed of 40 rpm at a temperature of 37 °C to simulate the gastric medium. One set of formulations (502H, 503H, 504H, CS-502H, CS-503H, CS-504H) was prepared and dispersed in trehalose solution at 5% (*w*/*v*) concentration. Then, 1 mL of this dispersion was added to solutions incubated at 37 ± 1 °C. Samples were collected after preincubation periods of 3, 9, and 24 h and centrifuged at 4.000 rpm for 5 min to precipitate NPs. Finally, the average PS of the NPs was determined [5,7]. This work was performed on fresh nanoparticle batches.

#### 2.3.4. High Performance Liquid Chromatography (HPLC) Conditions

The amount of CLR loaded into NPs and dissolution study of each formulation was performed using HPLC (Shimadzu Corporation, Kyoto, Japan) with reversed-phase Inert Sustain C_18_ (5.0 μm, 150 mm × 4.6 mm, GL Sciences Inc., Torrance, CA, USA) column. In the HPLC system, the mobile phase was acetonitrile: 0.035 M potassium dihydrogen phosphate buffer (KH_2_PO_4_) (55:45 *v*/*v*) mixture, the flow rate was 1 mL.min^−1^, while detection was performed at 200 nm at 30 °C. Injection volume was 25 μL. Mobile phase was prepared daily, degassed by sonication, and filtered through 0.45 μm membrane filter before the experiment. The method was validated for precision, accuracy, specificity, and linearity [23].

#### 2.3.5. Entrapment Efficiency

For entrapment efficiency, the method of extraction of CLR from NPs was used [5]. Accurately weighed (5 mg) NPs were dissolved in ethyl acetate. The resultant samples were filtered through 0.45 μm membrane filters and analyzed using HPLC. The entrapment efficiency (EE%) of NPs was calculated by Equation (1) [7].
(1)$$\mathrm{EE}\%=\left[\frac{\mathrm{Actual}\;\mathrm{amount}\;\mathrm{of}\;\mathrm{CLR}\;\mathrm{loaded}\;\mathrm{in}\;\mathrm{NPs}}{\mathrm{Theoretical}\;\mathrm{amount}\;\mathrm{of}\;\mathrm{CLR}\;\mathrm{loaded}\;\mathrm{in}\;\mathrm{NPs}}\right]\times 100.$$

#### 2.3.6. Dissolution and Release Kinetic Evaluation

*In vitro* dissolution study was performed in intestinal fluid phosphate buffer pH 6.8 containing 1% Tween^®^ 80 to retain sink conditions maintained at 37 ± 1 °C and 50 rpm using a USP Type II dissolution apparatus (PTWS 820D Pharma Test USP/EP Dissolution Testing Instrument, Hainburg, Germany). Pure CLR (5 mg) and equivalent NP formulations (equivalent 5 mg CLR) were placed in a cellulose acetate dialysis bag (dialysis tubing cellulose membrane, molecular weight cut-off: 14.000, Sigma-Aldrich, USA). After the addition of 1 mL of dissolution medium, the bag was sealed at both ends with special clamps for dissolution. After all the dialysis bags were closed, they were all immersed in the 500 mL of dissolution medium at the same time. Samples of the medium (5 mL) were withdrawn and replaced with fresh medium at 1, 3, 6, 9, 12, 24, 48, 72, 96, 120, and 144 h [24,25]. CLR concentration in the samples was analyzed by HPLC. The dissolution study was repeated three times for all NP formulations and pure CLR and the results were calculated as mean ± SD. The results were then plotted as cumulative release.

Data obtained in the *in vitro* dissolution studies were further investigated for release kinetics using DDSolver software program. DDSolver computer program was used to shorten the calculation time, eliminate calculation errors, and determine the correct release profile [26]. DDSolver software program was used for evaluating Higuchi, Korsmeyer-Peppas, Baker-Lonsdale, Peppas-Sahlin, and Weibull models.

### 2.4. Solid State Characterization of Nanoparticles

#### 2.4.1. Thermal (DSC) Analysis

The physical states of NPs were characterized by differential scanning calorimetry (DSC) (DSC-60, Shimadzu Scientific Instruments, Columbia, MI, USA). Aluminum crucibles with 3 mg samples were analyzed under nitrogen gas (50 mL.min^−1^) and heating rate of 10 °C.min^−1^ at a temperature range of 30 and 300 °C. Pure CLR and blank formulation were also analyzed and were used as references.

#### 2.4.2. FT-IR Analysis

FT-IR spectra of NPs were recorded using Shimadzu IR Prestige-21 (Shimadzu Corporation, Kyoto, Japan) at the wavelength range of 4000–500 cm^−1^. Pure CLR and blank formulations were also analyzed and were used as references.

#### 2.4.3. ^1^H-NMR Analysis

^1^H-NMR analyses were performed using UltraShield^TM^ CPMAS NMR (Brucker, Rheinstetten, Germany). Samples were prepared by dissolving formulations in deuterated chloroform (CDCI_3_). Pure CLR and blank formulations were also analyzed and were used as references.

### 2.5. Antimicrobial Activity Test

#### 2.5.1. Microorganisms

The following organisms were used in this study: *Staphylococcus aureus* (ATCC 25923), *Enterococcus faecalis* (ATCC 29212), *Listeria monocytogenes* (ATCC 1911), *Klebsiella pneumoniae* (ATCC 700603). All the bacterial strains were obtained from the ATCC (Rockville, MD, USA). The bacteria were in liquid nutrient broth (Merck) for fresh pure cultures.

#### 2.5.2. Inoculum

The standardization of the bacterial cell number used for susceptibility testing is of critical importance for obtaining accurate and reproducible results. The recommended final inoculum size for broth dilution is 5 × 10^5^ colony-forming units (CFU) per mL. For that reason, all inocula were set in 0.5 McFarland standard with McFarland Tube Densitometer for accurate and reproducible results.

#### 2.5.3. Broth Microdilution Method

Derivatives of substances were dissolved/suspended in DMSO and concentrations were prepared on ranging from 0.49 to 250 µL.mL^−1^. The prepared concentrations were distributed in duplicate 100 µl for each well on the 96-well plate. After that, fresh pure bacterial cultures, which were set in 0.5 McFarland standard in Mueller Hinton Broth (Sigma-Aldrich), were added at concentrations of 100 µL. After adding the bacterial cultures, final concentrations for all derivatives ranged from 250 to 0.49 µL.mL^−1^. At the end of this process, all plates were incubated for 24 h. All steps were performed as recommended by CLSI protocol.

### 2.6. Software Program

Microsoft Excel and DDSolver were employed for calculations.

## 3. Results and Discussion

### 3.1. Preparation of PLGA Nanoparticles and Surface Modification with Chitosan

PLGA is widely used in drug research due to the biocompatibility, biodegradability, and multidimensional degradation kinetics. The structural properties of PLGA, and especially its M_w_, affect many parameters in the properties of the NP formulations. The M_w_, in particular, affects the biological activity due to particle size, entrapment efficiency, release and release kinetics [27]. For this purpose, three different M_w_ of PLGA were used for prepared PLGA NPs. In order to investigate the effects of M_w_ on PLGA NP properties, three different formulations were prepared. The differences between the formulations were determined primarily by preparing formulations containing only Resomer^®^ RG 502 H (502H-coded formulations), only Resomer^®^ RG 503 H (503H-coded formulations), and only Resomer^®^ RG 504 H (504H-coded formulations).

The surface modification of the PLGA NPs was performed by CS, because the mucosal adhesive of CS had a positive ZP and CS interacted with negatively charged membranes and mucosa [15]. When the literature was examined, better results were obtained by surface modification with CS during NP preparation compared to after preparation [22]. Based on this literature information, we did not expose our prepared NPs to CS solution so as to avoid damage, we prepared them directly in CS solution. We modified the surface properties of all NPs prepared with different M_w_ of PLGA using CS (CS-502H, CS-503H, and CS-504H coded formulations). NP properties were examined in detail in terms of pharmaceutical nanotechnology, followed by antibacterial activity for all formulations.

### 3.2. Characterization of nanoparticles

#### 3.2.1. Particle Size, Polydispersity Index, Zeta Potential

PS, PDI, and ZP results are given in Table 2. The average PS of blank PLGA NPs varied between 142.0 nm and 154.4 nm, whereas the average PS of CLR-loaded PLGA NPs varied between 178.7 nm and 198.9 nm. The first thing that stands out in PS results is that PS increases with CLR loading. When a previous study with PLGA was reviewed, it was reported that PS increased when the active drug substance was loaded into NPs [5]. Another noteworthy point is that the M_w_ of PLGA affects PS results. When PS results were examined, the highest PS was obtained in the NP prepared with the lowest M_w_ of PLGA and the lowest PS was obtained in the NP prepared with the highest M_w_ of PLGA. The PS of NPs prepared with ResomerRG’s decreased with increase in M_w_ (M_w_ 502 H < M_w_ 503 H < M_w_ 504 H). These PS results can be explained by the hydrophobicity of the PLGA employed. At high M_w_, higher hydrophobicity and smaller PS were obtained due to longer aliphatic chains [28,29]. It was found that the addition of CS led to an increase in PS. PSs of NPs prepared with CS increased more than twofold compared to those prepared with PLGA alone. This may be explained due to CS-related viscosity increase, which reduces the shear stress during mixing of the emulsion on the magnetic stirrer and then leads to an increase in the PS of the emulsion droplets [15]. Similar results were obtained in literature [28,30]. Nonetheless, surface modification of NPs with hydrophilic components like CS is expected to improve their cellular uptake, as well as avoid the opsonization process, regardless of the reported size increase [28]. For the oral administration of CLR-loaded NPs, all NPs have desirable PS according to literature [31]. NPs prepared in this study were able to reach microcirculation by the blood capillaries or penetrate through pores present in the surfaces and membranes [15,31].

The PDI, which is a ratio that gives information about the homogeneity of the PS distribution in a given system, reflects the quality of the NP dispersion within the range of 0.0–1.0. PDI values ≤0.1 indicate the highest quality of dispersion. Most researchers recognize PDI values ≤0.3 as optimum values; however, values ≤0.5 are also acceptable [32]. According to literature, quality and monodisperse PLGA NPs and CS-coated PLGA NPs have been prepared.

When Table 2 is examined, negative ZP values are observed in uncoated PLGA NPs. ZP was observed in the range of −26.4 to −36.7 mV and −31.0 to −33.5 mV, respectively, in blank PLGA NPs and CLR-loaded NPs. PLGA in neutral medium has negative surface potential, attributed to the terminal carboxyl groups, and this could be verified by the ZP of negative obtained in uncoated PLGA NPs [15]. A colloidal system having ±30 mV as the ZP value is considered a stable formulation if dispersed as a colloidal dispersion in a liquid [33]. ZPs between −5.0 and −15.0 mV are in the limited flocculation zone; and the maximum flocculation zone between −5.0 and −3.0 mV was reported earlier [34]. When all the results are examined, it is seen that ZP values aren’t at the limit of flocculation. This shows the stability of all PLGA NPs prepared.

In surface-modified formulations with CS, the ZP of the NPs increased significantly to positive values. ZP was observed in the range of +77.1 to +77.5 mV and +75.0 to +76.6 mV, respectively, in blank CS-coated PLGA NPs and CS-coated CLR-loaded PLGA NPs. This is a consequence of the amino groups present in this polysaccharide and suggests that the PLGA NPs were adequately coated by CS [15]. Besides the high stability promoted by this high ZP value, positive surface charges are attracted by the negatively charged cell membranes, promoting adhesion and retention of the system at the site of action and in the intestinal epithelium, as well as increasing the absorption of the nanometric system [35]. In addition, the presence of CS may decrease the absorption and interaction of NPs by phagocytes because this absorption occurs more frequently on hydrophobic and negatively charged surfaces [36].

#### 3.2.2. Assessment of Cryoprotectant Effect on Nanoparticles

Figure 2 shows the effect of cryoprotectant addition on CLR-loaded PLGA NPs and CS-coated CLR-loaded PLGA NPs sizes. Freeze-drying (lyophilization) is a popular and preferred procedure for increasing the stability of various pharmaceutical products. Since NPs may increase in PS during freezing and drying steps, special agents must be added to the suspension before freezing to protect them. The most popular cryoprotectants for freeze-dried NPs are sugar derivatives, e.g., trehalose, sucrose, glucose, and mannitol [5]. In this study, trehalose was used at different concentrations and the PSs of NPs were measured after lyophilization to determine the optimum concentration of the cryoprotectant. Usually, the level of cryoprotectant used ranges from 0% to 50% by weight, which is solubilized directly in the NP suspension immediately prior to lyophilization. According to Figure 2, PSs were similar for trehalose concentrations added at 450 μL and 600 μL and are almost equal to fresh formulation. Both PLGA and CS-coated PLGA NP size was increased at low concentration of trehalose and therefore, 450 μL trehalose solution was found to be adequate for keeping the PS constant after freeze-drying.

#### 3.2.3. Evaluation of Gastrointestinal Stability of Nanoparticles

For better understanding of in vitro release profiles, stabilities of both PLGA NPs and CS-coated PLGA NPs were analyzed at 37 °C in different gastrointestinal media. Figure 3 shows NP sizes in pH 1.2 solution; intestinal fluid phosphate buffer solution (pH 6.8); phosphate buffer solution (pH 7.4); and distilled water. NPs composed of hydrolytic degradable polymers are known to degrade over time. It was reported that temperature and pH have very significant effects on long-term stability of drugs [7]. After incubation for 24 h at 37 °C in intestinal fluid phosphate buffer solution (pH 6.8), phosphate buffer solution (pH 7.4), and distilled water, average PS of NPs did not change significantly, when compared to the first PS measured for uncoated PLGA NPs. The results obtained indicate slow degradation of NPs in these medium at 37 °C, which also provides preliminary information for selection of dissolution medium. On the other hand, mean PS in pH 1.2 solution was determined to increase significantly in comparison to zero-time values. This indicates rapid degradation in the pH 1.2 solution. After incubation for 24 h at 37 °C in pH 1.2 solution; intestinal fluid phosphate buffer solution (pH 6.8); phosphate buffer solution (pH 7.4); and distilled water, average PS of CS-coated PLGA NPs did not change significantly, when compared to the zero-time PS measured. The results obtained indicate slow degradation of CS coated PLGA NPs in all media at 37 °C, which also provides preliminary information for selection of dissolution medium. As a result, it can be said that CS coating slows down the degradation process and protects NPs [5,7].

#### 3.2.4. High-Performance Liquid Chromatography (HPLC) Conditions

The HPLC method used in the study was validated for precision, accuracy and linearity [37,38]. Linearity was at a concentration range of 20–70 μg.mL^−1^. The method for CLR was precise due to RSD values of <2% for repeatability and intermediate precision. Recovery of the method was satisfactory owing to <2% RSD value. The CLR showed a linearity of y = 668.9238x + 2539.5397 (r^2^ = 0.999), and accuracy of 99.516% ± 0.543%, 99.713% ± 0.697%, and 99.605% ± 1.386% (mean% ± SD) for 45, 55 and 65 μg.mL^−1^, respectively (*n* = 6). The limit of detection (LOD) was 0.0014 μg.mL^−1^, while the limit of quantitation (LOQ) was 0.0042 μg.mL^−1^. The proposed procedure can be used for routine, simultaneous, and concurrent determination of CLR [39].

#### 3.2.5. Entrapment Efficiency

The corresponding data are shown in Table 2. Values of EE% for PLGA NPs ranged between 74% and 85%. The high values of encapsulation for PLGA NPs achieved are probably due to the lipophilic nature of the CLR, which presents low affinity to water phases and thus tends to migrate to the organic phase [21]. Another finding that draws attention to the results of EE% is that the results of the EE% were found to be as 502H > 503H > 504H. So, PLGA NPs prepared with Resomer^®^ RG 502 H had the highest EE%, whereas the NPs prepared with Resomer^®^ RG 504 H had the lowest EE% in PS results. This is explained in the literature by the loading of hydrophobic drugs to NPs being closely matched to the associated solid-state drug-polymer solubility, i.e., the ability of the polymeric matrix to keep the drug dispersed. This is briefly summarized as follows: the solid-state solubility of the drug in the polymer was increased with a decrease in M_w_ [28,40]. This theory supports the EE% results we obtained in this study. The EE% of CLR in CS coated-PLGA NPs ranged between 62% and 63%. A decrease in EE% was observed in CS-coated PLGA NPs compared with uncoated PLGA NPs. When the cause of this condition was investigated in the literature, similar results were obtained in CS-coated paclitaxel-loaded PLGA NPs [41]. This can be explained briefly, as a possible cause of CS hydrophilicity, which may prevent entrapment of the hydrophobic drug.

#### 3.2.6. Dissolution and Release Kinetic Evaluation

We evaluated the amount of CLR released from developed PLGA NPs, CS-coated PLGA NPs, and CLR suspension as a function of time. *In vitro* dissolution profiles of the pure CLR, uncoated and CS-coated PLGA NP formulations prepared are presented in Figure 4. Twenty-four hour dissolution profiles are also presented in Figure 4b to provide a more detailed view. The *in vitro* dissolution profiles of the uncoated and coated PLGA NPs showed a similar biphasic configuration, consisting of initial burst followed by a sustained release. The release of CLR from all NPs continued over 144 h while pure CLR exhibited a rapid release of 94.1% ± 3.8% (mean ± SD) in 3 h. The dissolution rates observed from 502H, 503H, 504H, CS-502H, CS-503H and CS-504H coded NP formulations after 144 h were 96.3% ± 2.2%, 87.2% ± 4.6%, 58.2% ± 3.2%, 88.4% ± 1.1%, 76.4% ± 5.1%, and 47.5% ± 6.4%, respectively, demonstrating extended release from all formulations, relative to pure CLR. Figure 4 shows that in the case of CS-coated PLGA NPs, the CLR release rate was slightly slower than that of the uncoated PLGA NPs. CLR released was primarily due to desorption and diffusion of the drug from the surface and the small pores on the surface of the NPs. In this case, the slower CLR release rate might be attributed to the variation in desorption and diffusion of drug because of CS coating on the PLGA NPs [42]. The release results indicate that the CS -NPs exhibited a CLR release pattern similar to the uncoated NPs, but with a slower CLR release rate.

When the results of the dissolution study were examined, a rapid release was observed up to 24 h. Cumulative release rates with burst release at the 24th hour of 502H, 503H, 504H, CS-502H, CS-503H and CS-504H coded formulations were found as follows; 76.4% ± 3.6%, 66.1% ± 2.8%, 38.2% ± 3.4%, 54.6% ± 4.2%, 48.8% ± 2.8%, and 31.4% ± 2.4, respectively. The rapid release of these formulations was finished at the end of the 24th hour and passed to their second phase, which was slow release. The most important factor that draws attention here is that the release rates in the first phase (burst effect) were found for uncoated formulation as 502H > 503H > 504H and for CS-coated formulation as CS-502H > CS-503H > CS-504H and a similar result was observed at the end of 144 h. There seems to be an inverse relationship between amount of release rate and PS. It was reported previously that large NPs degrade faster than small NPs. This is probably due to the increased accumulation of acidic products during polymer hydrolysis in large NPs where hydrolysis starts immediately in PLGA systems [7,43]. In this case, it can be said that the dissolution rates are indirectly influenced by M_w_. This hypothesis is supported by the fact that the 504H-coded formulation with small PS has the least dissolution rate and 502H-coded formulation with largest PS has the highest dissolution rate.

Kinetic modeling of CLR release from NPs is shown in Table 3, Table 4, Table 5, Table 6 and Table 7. After obtaining the release profiles, data were transferred to the DDSolver program to determine the four most important and popular criteria: coefficient of determination (Rsqr, R^2^, or COD), adjusted coefficient of determination (Rsqr_adj or R^2^_adjusted_), Akaike information criterion (AIC), and model selection criterion (MSC). The highest R^2^, R^2^_adjusted_, and MSC values and the lowest AIC values were used for evaluating Higuchi, Korsmeyer–Peppas, Baker–Lonsdale, Peppas–Sahlin and Weibull models [26]. In the investigation of drug release kinetics, it was seen that most of the studies were applied to the mathematical models for the total duration of the release [3,5,7]. This is known to be the right approach, but when new approaches are examined, the burst release kinetics of drugs and the total release kinetics can vary [44]. Therefore, in this study, a burst effect of 24 h was observed when the *in vitro* dissolution results were examined, and both the burst release kinetics and the 144 h kinetics were examined. Burst release leads to a high level of drug delivery at first and it is also important for the drug delivery system to provide a therapeutic concentration for effective treatment. Since the burst effect occurred in a very short time compared to the whole release time, most published results did not specifically investigate burst effect [45]. The R^2^ and R^2^_adjusted_ values for the Higuchi, Korsmeyer–Peppas, and Baker–Lonsdale models were smaller than other models for all NPs, which were relatively small. This suggests the drug release does not comply with Higuchi, Korsmeyer–Peppas, Baker–Lonsdale models. When the kinetics results of 24 h and 144 h were examined, the values of R^2^, R^2^_adjusted_, MSC, and AIC were very similar for Peppas–Sahlin and Weibull models (Table 6 and Table 7). In other words, higher correlation was observed in the Peppas–Sahlin model and Weibull model. Therefore, results of this study indicate that release of CLR from NPs is not predominantly driven by a solo mechanism, but a combined mechanism of Fickian (pure diffusion phenomenon) and non-Fickian release (due to the relaxation of the polymer chain between networks). When the literature is examined, similar results were encountered [46]. As seen in the results, the kinetic was not affected directly by M_W_ or affected by the natural structure of the PLGA and CS.

### 3.3. Solid State Characterization of Nanoparticles

#### 3.3.1. Thermal (DSC) Analysis

DSC curves/thermograms of intact CLR, blank NPs and CLR-loaded NPs are demonstrated in Figure 5. CLR (Figure 5a) presented a melting endothermic peak at 228.84 °C. The result was found to be appropriate compared to the literature [24]. No clear peak presented (Figure 5h–m) in the thermograms of the prepared uncoated and CS-coated PLGA NPs due to any possible decrease in the drug crystallinity and/or solvation of the drug in the melted carrier and/or heat-induced interaction between drug and polymer [47]. This disappearance of melting peaks of CLR from the NP formulation thermograms indicated that CLR was likely encapsulated in the amorphous state and molecularly dispersed in the polymeric structure [48,49]. The thermograms demonstrated that there was no interaction between the CLR and the polymers. These data are important because the presence of a drug in molecular dispersion form helps in its sustained release property [50].

#### 3.3.2. FT-IR Analysis

FT-IR spectrum of CLR and formulations are given in Figure 6. FT-IR spectrum of CLR (Figure 6a) showed peak for C–H stretching vibration at 2978 cm^−1^; peak of -O–C=O stretching vibration in lactone ring and –C=O stretching ketone group at 1732 cm^−1^ and around 1600 cm^−1^, respectively, peaks at 1166 cm^−1^ and 1010 cm^−1^ refer to -O-ether functional band. The FT-IR spectrum of CLR was consistent with the literature [24]. Resomer^®^ RG 502 H, Resomer^®^ RG 503 H, and Resomer^®^ RG 502 H are PLGA polymers having the same chemical structures but with different M_w_. Carbonyl groups (C=O) observed between 1749–1755 cm^−1^ with intense bands were attributed to stretching vibration present in both monomers, while medium density bands between 1300 and 1100 cm^–1^ were, respectively, attributed to asymmetric and symmetric C–C(=O)-–O stretches in blank formulation spectrum (Figure 6b–g) [5,7]. FT-IR spectrum of CS-coated blank formulation (Figure 6e–g) showed characteristic absorption bands around 3750–2500 cm^−1^, which represent –OH, -CH_2_, and -CH_3_ aliphatic groups, and bands at around 1400–1500 cm^−1^ that represent –NH group bending vibration and vibrations of the -OH group of the primary alcohol, respectively. Since the PLGA peaks are overly dominant, they have reduced the selectivity of the CS peaks, but in some regions the peaks of CS suppressed the peak peaks of the PLGA. This can be considered as an indicator of CS coating [49]. The FT-IR spectrum obtained for CLR loaded all NPs that displayed the typical bands of drug and PLGA; a possible new carbonyl stretching band at higher wave number (due to potential interactions between CLR and all type of PLGA) was masked by the broad carbonyl stretching band at 1753–1751 cm^–1^ (also seen in the all blank formulation spectrum). Similar results were observed with the flurbiprofen-loaded Resomer^®^ RG 756 NP and the flurbiprofen-loaded poly-ε-capro-lactone NP [51,52]. Distinctive peaks of CLR were not seen in the spectra of all CLR-loaded NP formulations indicating the molecular dispersion of CLR in the polymeric matrices, which was supported by the DSC results [49]. The absence of CLR distinctive peaks confirmed encapsulation of drug within the polymeric structure.

#### 3.3.3. ^1^H-NMR Analysis

In the ^1^H-NMR analysis, the active substance and polymer signals can be better observed than in the FT-IR analysis as the materials are dissolved in the deutero-solvent. The ^1^H-NMR spectra of CLR and formulation are shown in Figure 7. The ^1^H-NMR spectrum of CLR (Figure 7a) shows a similarity to the literature [53]. Characteristic ^1^H-NMR signals for CLR are dense signals of 1–5 ppm. PLGA contains two types of structural units: the most intense signals of CH (5.1 ppm) and CH_3_ (1.4 ppm) from lactic acid and CH_2_ (4.8 ppm) from glycolic acid. The CH_2_ hydrogens were diastrophic and divided into two, ending in two pairs of peaks that were assigned for both protons. Higher CH_2_, CH_3_, and CH signals in the copolymer caused those peaks to expand. When the ^1^H-NMR spectrum of both blank and CLR-loaded NPs is examined, specific peaks of the PLGA polymer are observed in Figure 7b–m. Signals at 4.9 ppm are for D-glucosamine unit, signals at 4.7 ppm for N-acetyl-D-glucosamine, signals at 3.2 ppm and 2.1 ppm for -CH3 are specific signals for chitosan [54]. When the ^1^H-NMR spectrum of both blank and CLR-loaded NPs is examined, specific peaks of the CS are observed in Figure 7e–m. Figure 7h–m shows specific ^1^H-NMR signals of CLR in both PLGA NPs and CS-coated PLGA NPs. This indicates that the CLR has been successfully loaded into the formulations [55].

### 3.4. Antimicrobial Activity Test Results

The antimicrobial activities of prepared compounds were tested on microorganisms including *Staphylococcus aureus* (ATCC 25923), *Enterococcus faecalis* (ATCC 29212), *Listeria monocytogenes* (ATCC 1911), and *Klebsiella pneumoniae* (ATCC 700603). The results are given in Table 8.

According to the antibacterial activity test results, obtained from minimum inhibitory concentration tests (MIC), many of the compounds were found to be active against the identified microorganisms compared to control. All formulations exhibited remarkable antibacterial activity, especially on *Staphylococcus aureus* (ATCC 25923), *Listeria monocytogenes* (ATCC 1911), and *Klebsiella pneumoniae* (ATCC 700603). 503H, 504H, and CS-504H showed significant activity compared to standard drug (Chloramphenicol) and CLR against *Staphylococcus aureus* (ATCC 25923). Only one of four microorganisms, *Enterococcus faecalis* (ATCC 29212), has not been found to be susceptible for these formulations.

Antimicrobial effects can be discussed in terms of pharmaceutical technology as follows. In this study, PLGA with three different M_w_ and CS were used. When PSs were examined, the highest PS was found in the CS-504H-coded formulation. Compared to non-CS-coated formulations, the increase in PS was most observed in the CS-504 H-coded formulation. This is proof that CS is more present in the CS 504H-coded formulation than CS-502H and CS 503H. When the literature is examined, there is a direct correlation between the M_w_ of chitosan and the antimicrobial effect. Many studies in the literature have shown that as the M_w_ of chitosan increases, its antimicrobial activity increases [56,57]. When the results were discussed from a different perspective, according to these literature, the presence of more CS in CS-504H increased the antimicrobial efficacy on *Staphylococcus aureus*. The other two formulations with high antimicrobial effect on *Staphylococcus aureus* are formulations coded with 503H and 504H. When the literature is examined, it has been observed in many studies that the antimicrobial effect increases when the PS decreases [58,59]. When the results of PS are examined, the two formulations having the lowest PS were 503H and 504H coded formulations. It can be said that the low PS increased the antimicrobial effect on *Staphylococcus aureus*. There have been studies in the literature with CLR [24,60]. When these studies were examined, the MIC value of CLR against *Staphylococcus aureus* was found to be 0.5 µg.mL^−1^ and 12.0 µg.mL^−1^. In this study, the MIC value of the CLR against *Staphylococcus aureus* was observed to be 125 µg.mL^−1^. While the bacterial chain used in our study was ATCC 25923 for *Staphylococcus aureus*, the bacterial chains in other studies were ATCC 29213 and MTCC86 for *Staphylococcus aureus.* These are two different strains (ATCC 29213 and MTCC86) of *Staphylococcus aureus*. The same species of bacteria with different strain designations codes have different sensitivity to antimicrobial drugs [24]. This reason explains the differences in the literature with the results we obtained.

As a result, all the formulations we prepared have increased antimicrobial effect compared to CLR. Due to the fact that free drug and NPs were investigated using the same concentration of CLR, the improvement in antimicrobial effect activity could be due to better penetration of the NPs into bacterial cells and better delivery of CLR to its site of action [61]. NPs are capable of being endocytosed by phagocytic cells and releasing the drug into those cells [62]. The CLR-loaded NPs could be suitable for delivery of CLR to phagocytic cells to achieve better treatment of infection compared with treatment using free CLR. This indicates that the newly designed antibiotic-releasing NPs may be appropriate for antimicrobial treatment.

## 4. Conclusions

In this study, nanoparticles were prepared with three different M_w_ of PLGA and chitosan. We have successfully formulated clarithromycin-loaded nanoparticles by nanoprecipitation technique. Detailed characterizations were made to all formulations and obtained results showed that the M_w_ of PLGA and chitosan modification exerted significant influence on the nanoparticle properties. The particle size of the prepared nanoparticles was in the range of 178 to 578 nm and it was affected by the M_w_ and chitosan coating. In surface-modified formulations with chitosan, the zeta potential of the nanoparticles increased significantly to positive values. The entrapment efficiency results of formulations intended for oral use are highly acceptable. Entrapment efficiency varied from 62% to 85%, depending upon the M_w_ and chitosan coating. In vitro release studies of clarithromycin-loaded nanoparticles showed an extended release up to 144 h. Peppas–Sahlin and Weibull kinetic model was found to fit best for clarithromycin release from nanoparticles. Solid-state characterization by DSC, FT-IR, and ^1^H-NMR has proven successful in the production of nanoparticles. The 503H, 504H and CS-504H coded formulations exhibited remarkable antibacterial activity, especially on *Staphylococcus aureus*. At least in terms of some formulations, improved pharmacodynamic effects were also obtained from antibacterial activity studies and it could be concluded that clarithromycin-loaded nanoparticles seem to be a promising delivery system for the oral application.

## Figures and Tables

**Figure 1 polymers-11-01632-f001:**
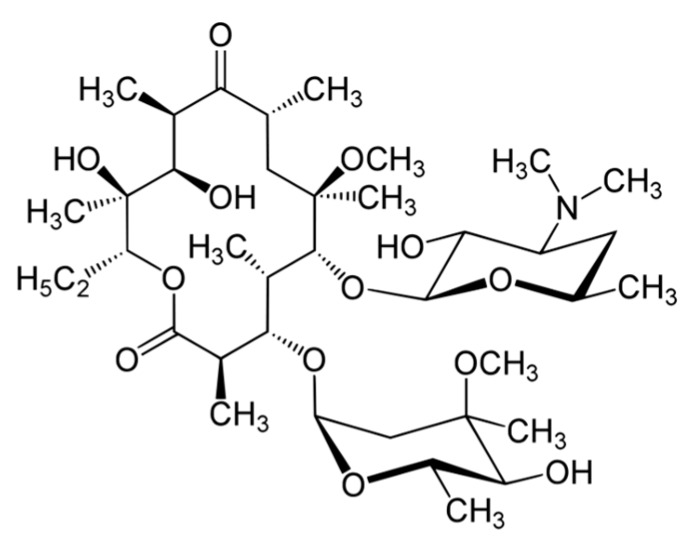
Chemical structure of clarithromycin (CLR).

**Figure 2 polymers-11-01632-f002:**
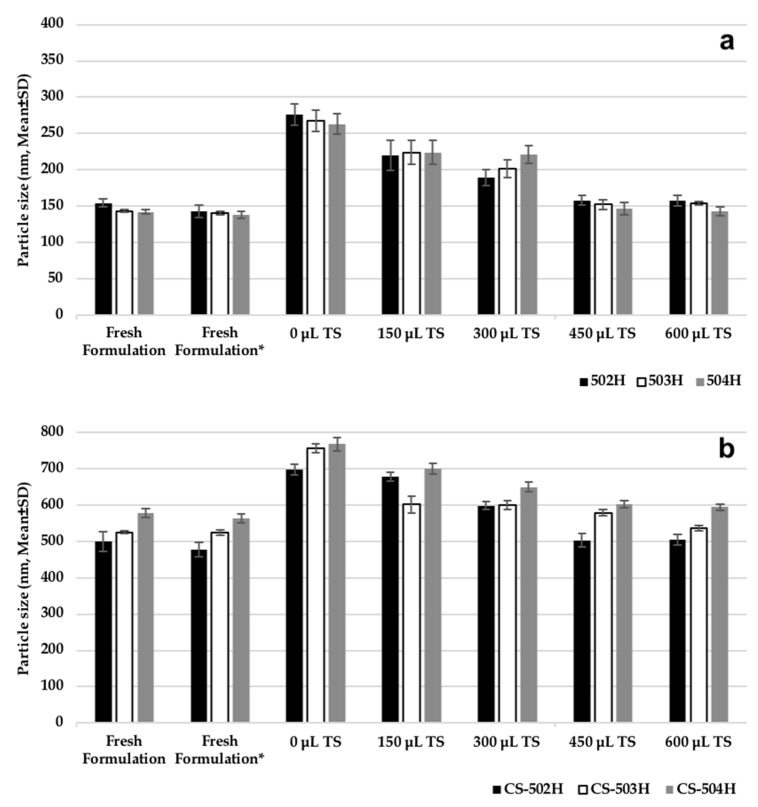
Effect of cryoprotectant on particle size of nanoparticles prepared; (**a**) PLGA nanoparticles (NPs), (**b**) chitosan-coated PLGA NPs. *: Fresh formulation after centrifugation and dispersing in 1 mL trehalose solution (5% *w*/*v*).

**Figure 3 polymers-11-01632-f003:**
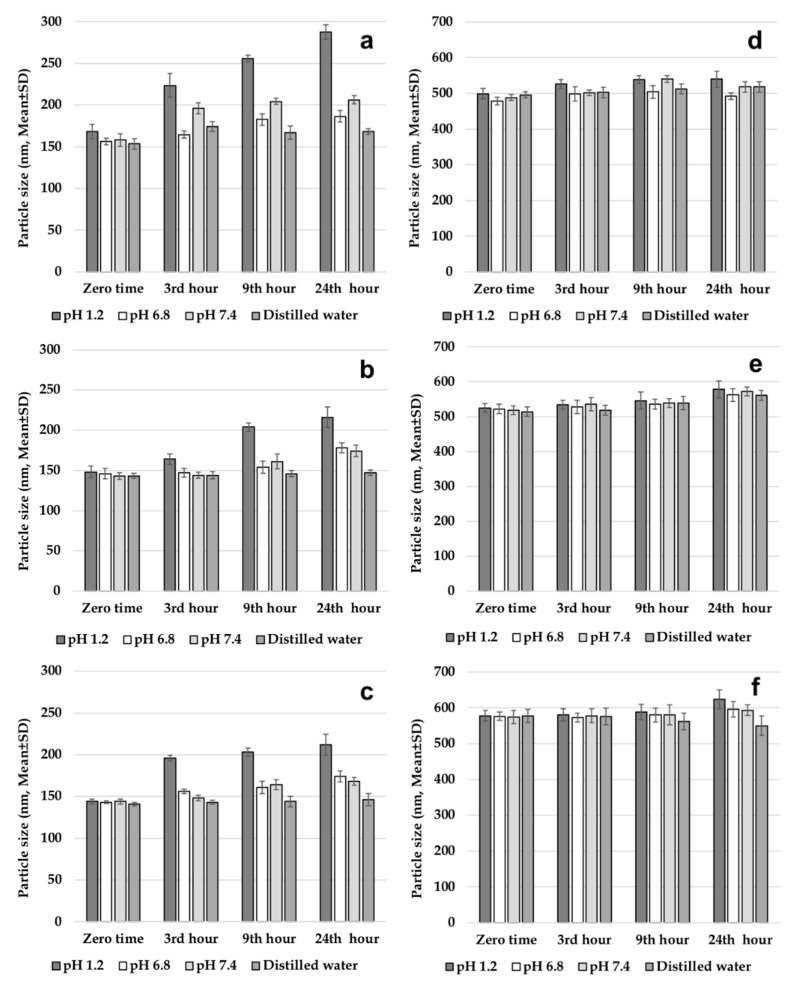
Gastrointestinal stability of nanoparticles, (**a**) 502H, (**b**) 503H, (**c**) 504H, (**d**) CS-502H, (**e**) CS-503H, (**f**) CS-504H.

**Figure 4 polymers-11-01632-f004:**
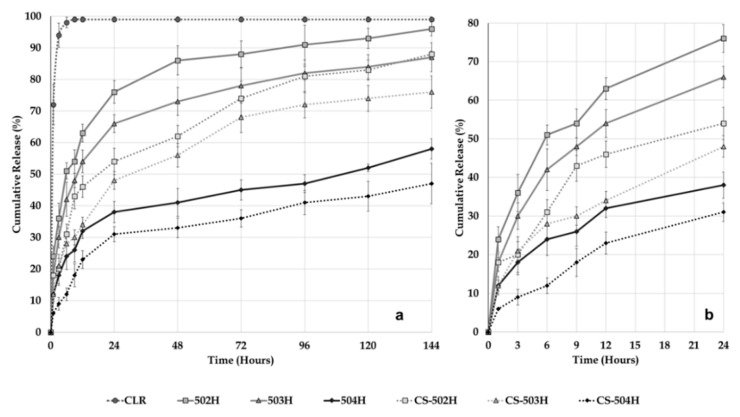
*In vitro* release of pure CLR, CLR from PLGA NPs and CS-coated PLGA NPs at 37 °C in intestinal fluid phosphate buffer pH 6.8 supplemented with 1% Tween 80. (**a**) 144 hours release profile, (**b**) 24 hours release profile.

**Figure 5 polymers-11-01632-f005:**
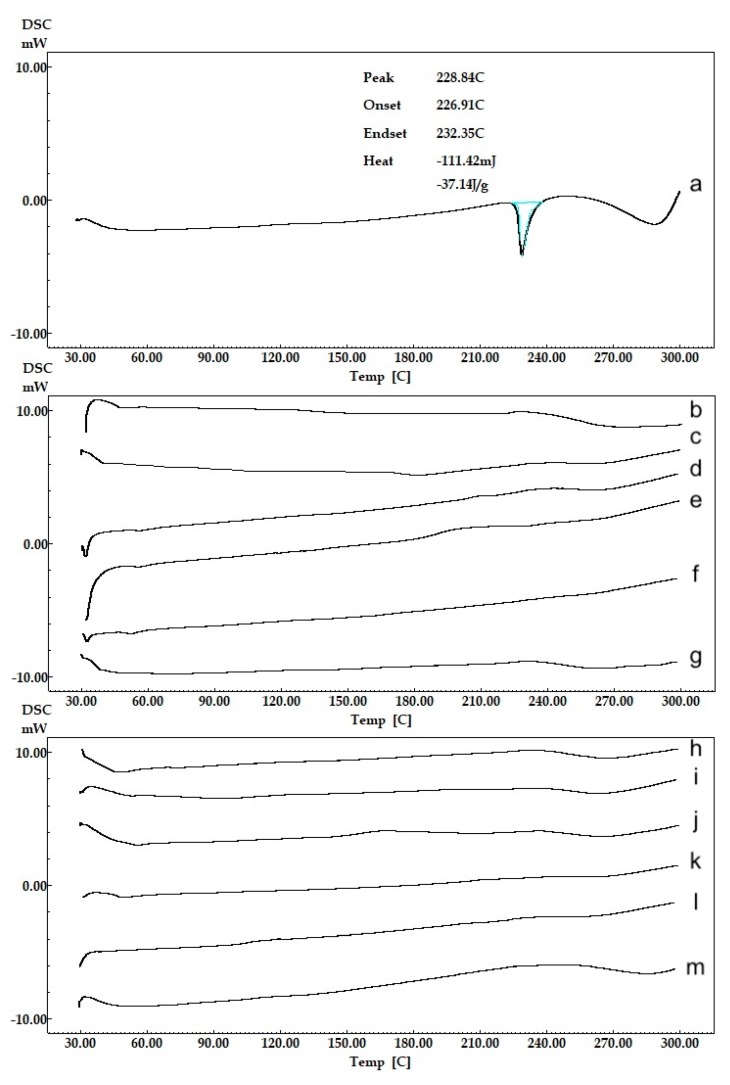
Thermal (DSC) analysis curves of CLR and NPs, (**a**) CLR, (**b**) 502H-Blank, (**c**) 503H-Blank, (**d**) 504H-Blank, (**e**) CS-502H-Blank, (**f**) CS-503H-Blank, (**g**) CS-504H-Blank, (**h**) 502H, (**i**) 503H, (**j**) 504H, (**k**) CS-502H, (**l**) CS-503H, (**m**) CS-504H.

**Figure 6 polymers-11-01632-f006:**
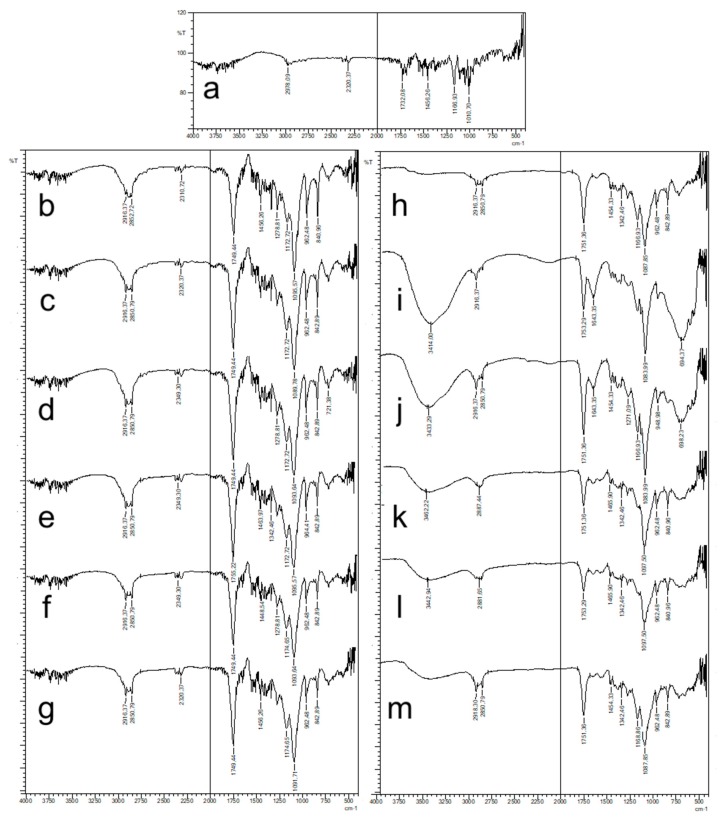
FT-IR spectrum of CLR and NPs, (**a**) CLR, (**b**) 502H-Blank, (**c**) 503H-Blank, (**d**) 504H-Blank, (**e**) CS-502H-Blank, (**f**) CS-503H-Blank, (**g**) CS-504H-Blank, (**h**) 502H, (**i**) 503H, (**j**) 504H, (**k**) CS-502H, (**l**) CS-503H, (**m**) CS-504H.

**Figure 7 polymers-11-01632-f007:**
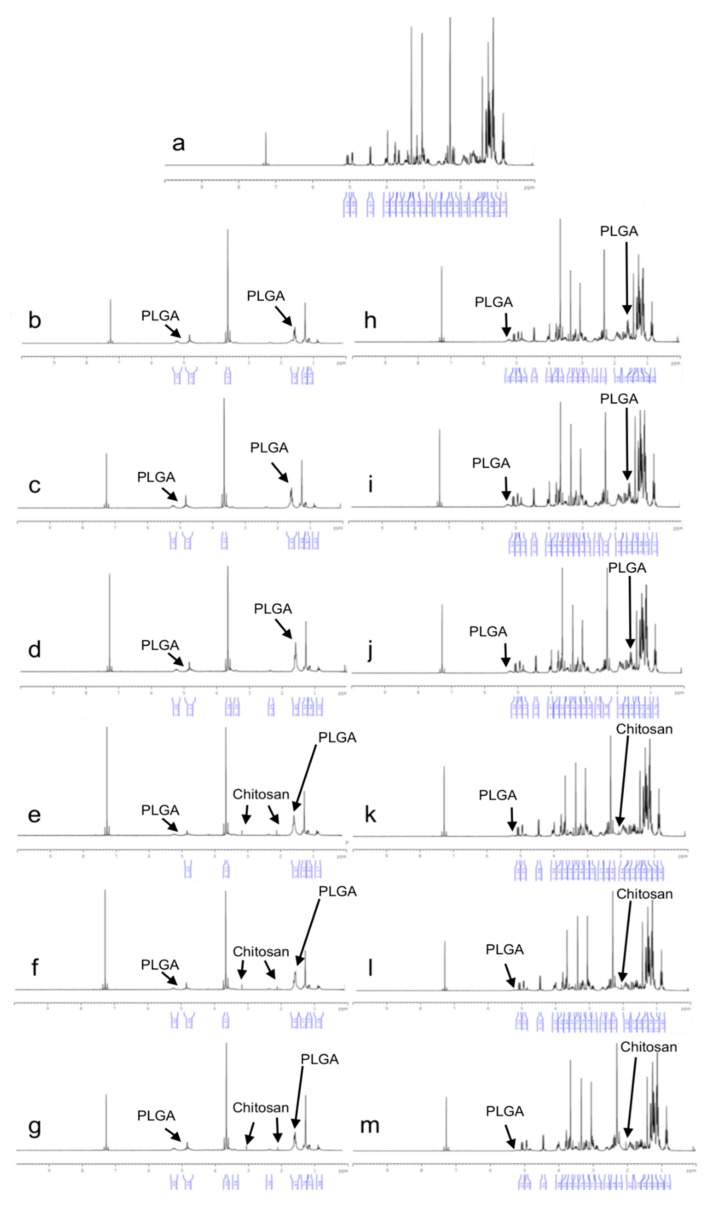
^1^H-NMR spectrum of CLR and NPs, (**a**) CLR, (**b**) 502H-Blank, (**c**) 503H-Blank, (**d**) 504H-Blank, (**e**) CS-502H-Blank, (**f**) CS-503H-Blank, (**g**) CS-504H-Blank, (**h**) 502H, (**i**) 503H, (**j**) 504H, (**k**) CS-502H, (**l**) CS-503H, (**m**) CS-504H.

**Table 1 polymers-11-01632-t001:** Formulation ingredients.

Code	502 H *	503 H *	504 H *	Span 60	ACN *	CLR *	Pluronic F-68 (0.5%)	Chitosan * (0.25 %)
502H-Blank	90 mg	-	-	30 mg	3 mL	-	10 mL	-
503H-Blank	-	90 mg	-	30 mg	3 mL	-	10 mL	-
504H-Blank	-	-	90 mg	30 mg	3 mL	-	10 mL	-
502H	90 mg	-	-	30 mg	3 mL	9 mg	10 mL	-
503H	-	90 mg	-	30 mg	3 mL	9 mg	10 mL	-
504H	-	-	90 mg	30 mg	3 mL	9 mg	10 mL	-
CS-502H-Blank	90 mg	-	-	30 mg	3 mL	-	-	10 mL
CS-503H-Blank	-	90 mg	-	30 mg	3 mL	-	-	10 mL
CS-504H-Blank	-	-	90 mg	30 mg	3 mL	-	-	10 mL
CS-502H	90 mg	-	-	30 mg	3 mL	9 mg	-	10 mL
CS-503H	-	90 mg	-	30 mg	3 mL	9 mg	-	10 mL
CS-504H	-	-	90 mg	30 mg	3 mL	9 mg	-	10 mL

* 502 H: Resomer RG 502 H, 503 H: Resomer RG 503 H, 504 H: Resomer RG 504 H, ACN: acetone, CLR: clarithromycin, Chitosan solution: including Pluronic F-68 (0.5%).

**Table 2 polymers-11-01632-t002:** Particle size (PS), polydispersity index (PDI), zeta potential (ZP), and entrapment efficiency (EE%) results.

Code	PS *	PDI *	ZP *	EE% *
502H-Blank	154.4 ± 5.4	0.154 ± 0.086	−26.4 ± 0.7	-
503H-Blank	143.3 ± 1.3	0.161 ± 0.024	−34.8 ± 1.5	-
504H-Blank	142.0 ± 2.6	0.160 ± 0.082	−36.7 ± 0.5	-
502H	198.9 ± 1.5	0.286 ± 0.067	−33.2 ± 1.8	85.199 ± 2.138
503H	191.3 ± 2.9	0.216 ± 0.008	−31.0 ± 1.6	77.836 ± 4.614
504H	178.7 ± 1.6	0.139 ± 0.060	−33.5 ± 1.2	74.627 ± 3.390
CS-502H-Blank	441.1 ± 7.9	0.300 ± 0.056	+75.1 ± 0.9	-
CS-503H-Blank	419.0 ± 9.9	0.314 ± 0.058	+77.5 ± 4.2	-
CS-504H-Blank	407.9 ± 3.6	0.334 ± 0.065	+71.1 ± 1.8	-
CS-502H	499.9 ± 7.4	0.311 ± 0.110	+75.0 ± 1.9	63.460 ± 3.910
CS-503H	525.2 ± 2.9	0.326 ± 0.120	+76.6 ± 1.65	62.771 ± 1.911
CS-504H	578.3 ± 6.8	0.321 ± 0.076	+75.8 ± 0.9	62.463 ± 0.026

* Results given as mean ± SD, PS: Particle size (nm), PDI: Polydispersity index, ZP: Zeta potential (mV), EE%: Entrapment efficiency (%).

**Table 3 polymers-11-01632-t003:** Kinetic results for Higuchi Model.

Model	Evaluation Criteria	Formulations											
		24 Hours Kinetics						144 Hours Kinetics					
		502H	503H	504H	CS-502H	CS-502H	CS-502H	502H	503H	504H	CS-502H	CS-502H	CS-502H
**Higuchi**	**R^2^**	0.938	0.960	0.947	0.943	0.980	0.976	0.569	0.667	0.762	0.860	0.904	0.910
	**R^2^_adjusted_**	0.938	0.960	0.947	0.943	0.980	0.976	0.569	0.667	0.762	0.860	0.904	0.910
	**AIC**	40.462	35.621	29.592	35.649	25.676	21.375	103.060	97.800	81.997	87.731	80.915	68.140
	**MSC**	1.843	2.347	2.016	2.021	3.069	3.109	0.199	0.509	0.876	1.481	1.889	1.985

**Table 4 polymers-11-01632-t004:** Kinetic results for Korsmeyer-Peppas Model.

Model	Evaluation Criteria	Formulations											
		24 Hours Kinetics						144 Hours Kinetics					
		502H	503H	504H	CS-502H	CS-502H	CS-502H	502H	503H	504H	CS-502H	CS-502H	CS-502H
**Korsmeyer-Peppas**	**R^2^**	0.993	0.990	0.994	0.969	0.994	0.982	0.982	0.982	0.991	0.992	0.994	0.982
	**R^2^_adjusted_**	0.991	0.988	0.993	0.962	0.993	0.978	0.957	0.955	0.980	0.983	0.985	0.956
	**AIC**	27.293	27.790	16.539	33.475	18.853	21.587	77.498	75.749	54.040	64.710	60.683	61.474
	**MSC**	3.725	3.466	3.881	2.331	4.043	3.079	2.329	2.347	3.205	3.399	3.575	2.540

**Table 5 polymers-11-01632-t005:** Kinetic results for Baker-Lonsdale Model.

Model	Evaluation Criteria	Formulations											
		24 Hours Kinetics						144 Hours Kinetics					
		502H	503H	504H	CS-502H	CS-502H	CS-502H	502H	503H	504H	CS-502H	CS-502H	CS-502H
**Baker-Lonsdale**	**R^2^**	0.962	0.971	0.945	0.941	0.987	0.961	0.687	0.861	0.833	0.950	0.957	0.916
	**R^2^_adjusted_**	0.962	0.971	0.945	0.941	0.987	0.961	0.687	0.861	0.833	0.950	0.957	0.916
	**AIC**	37.014	33.383	29.906	35.903	22.566	24.810	99.208	87.319	77.743	75.423	71.306	67.328
	**MSC**	2.336	2.667	1.971	1.984	3.513	2.619	0.520	1.383	1.230	2.507	2.690	2.052

**Table 6 polymers-11-01632-t006:** Kinetic results for Peppas-Sahlin Model.

Model	Evaluation Criteria	Formulations											
		24 Hours Kinetics						144 Hours Kinetics					
		502H	503H	504H	CS-502H	CS-502H	CS-502H	502H	503H	504H	CS-502H	CS-502H	CS-502H
**Peppas-Sahlin**	**R^2^**	0.997	0.998	0.995	0.971	0.994	0.985	0.990	0.992	0.984	0.988	0.997	0.980
	**R^2^_adjusted_**	0.995	0.996	0.992	0.957	0.991	0.978	0.988	0.990	0.980	0.985	0.996	0.975
	**AIC**	24.226	19.884	17.220	34.845	21.460	22.075	62.011	57.619	53.926	62.193	44.420	54.189
	**MSC**	4.163	4.596	3.783	2.136	3.671	3.009	3.620	3.857	3.215	3.609	4.930	3.147

**Table 7 polymers-11-01632-t007:** Kinetic results for Weibull Model.

Model	Evaluation Criteria	Formulations											
		24 Hours Kinetics						144 Hours Kinetics					
		502H	503H	504H	CS-502H	CS-502H	CS-502H	502H	503H	504H	CS-502H	CS-502H	CS-502H
**Weibull**	**R^2^**	0.993	0.999	0.993	0.963	0.988	0.970	0.997	0.997	0.987	0.987	0.995	0.977
	**R^2^_adjusted_**	0.989	0.998	0.989	0.944	0.982	0.956	0.996	0.996	0.985	0.984	0.994	0.972
	**AIC**	29.294	15.444	19.732	36.659	25.936	26.950	48.613	47.099	50.713	63.687	49.256	55.717
	**MSC**	3.439	5.230	3.424	1.876	3.031	2.313	4.736	4.734	3.483	3.485	4.527	3.020

**Table 8 polymers-11-01632-t008:** Antimicrobial test results.

Formulation Codes	*Staphylococcus aureus* (ATCC 25923)	*Enterococcus faecalis* (ATCC 29212)	*Listeria monocytogenes* (ATCC 1911)	*Klebsiella pneumoniae* (ATCC 700603)
CS-502H	62.50 µg.mL^−1^	250.00 µg.mL^−1^	31.25 µg.mL^−1^	62.50 µg.mL^−1^
502H	62.50 µg.mL^−1^	250.00 µg.mL^−1^	31.25 µg.mL^−1^	31.25 µg.mL^−1^
CS-503H	62.50 µg.mL^−1^	250.00 µg.mL^−1^	62.50 µg.mL^−1^	31.25 µg.mL^−1^
503H	31.25 µg.mL^−1^	250.00 µg.mL^−1^	62.50 µg.mL^−1^	31.25 µg.mL^−1^
CS-504H	31.25 µg.mL^−1^	250.00 µg.mL^−1^	62.50 µg.mL^−1^	31.25 µg.mL^−1^
504H	31.25 µg.mL^−1^	250.00 µg.mL^−1^	62.50 µg.mL^−1^	62.50 µg.mL^−1^
CLR	125.00 µg.mL^−1^	250.00 µg.mL^−1^	125.00 µg.mL^−1^	62.50 µg.mL^−1^
Control (Chloramphenicol)	7.81 µg.mL^−1^	15.625 µg.mL^−1^	15.625 µg.mL^−1^	7.81 µg.mL^−1^

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
