# Peer review of "Clarithromycin-Loaded Poly (Lactic-co-glycolic Acid) (PLGA) Nanoparticles for Oral Administration: Effect of Polymer Molecular Weight and Surface Modification with Chitosan on Formulation, Nanoparticle Characterization and Antibacterial Effects"

_polymers, 2019, doi:10.3390/polym11101632_

Round 1

Reviewer 1 Report

The manuscript by  Öztürk et al describes development, physiochemical characterization, and evaluation of poly (lactic-co-glycolic acid) (PLGA) NPs and chitosan (CS) coated PLGA NP for oral delivery of  antibiotics, Clarithromycin. The manuscript is well organized and most of the conclusions are supported with data. However, antibacerial properties of clarithromycin loaded PLGA NP is reported in literature (Valizadeh H et al 2012), author need to clearly mention and compare  how this report is different and antibacterial property need more experiments to support the conclusion. 

-Title is very long need to be short.

-Line 23 " also elucidate the effect of three molecular weight (Mw) of PLGA" does not make sense. elucidated instead of elucidate and  please mention which three molecular weights instead of weight.

-Line 32-34 needs sentence modification

L65-72 different font.

-Introduction needs what's known in literature as there are reports on CLR loaded PLGA including use of CS coating, and clearly state how this article fill if there is gap area.

-L139 need space between degree and C. Double check the degree symbol abd space through out the MS (L158)

-2.3.1 different font with shadow

-L224-227 all bacterial strain  name should be italic.

-L280 the value in the table suggest more than 2 fold increase in PS upon adding CS.

-L386 Twenty four instead of 24

DSC results it is surprising that the peak is completely disappeared in all formulations. I am not sure if the control was run in parallel and if the concentration used were OK because no peak could be result of methodological problem. Another way to obatain thermal stability of these polymer is Thermogravimetric analysis. Please perform TGA analysis to strengthen the prediction.   

-      FTIR figure 6 graph resolutions are very low and not clear (the numbers). Please take the value and use software like Origin to make a clear graph.

-Need to re-write the FT-IR results, there are some open questions- peak at 3400 is also present in fig6m why it was not indicated? Is this broader peak means anything because it is not present in blank samples. Do you consider only 2300 as CLR representative peak which you think disappeared after formulation?

L531-534 bacterial names should be in italic.

You can write this section better emphasizing on broad spectrum which means you have tested both gram positive and gram negative bacteria. These bacteria have different cell wall and generally gram positives are more susceptible.These values does not seems right in staph. and Klebsiella.

1 How come CS coating at 502H is less efficient 62 than 502H 31 for Kleb and similarly 503H for staph? please do disc diffusion assay(DDA) and CFU count to get actual number and compare if the results from MIC agree with DDA.

Author Response

Response to Reviewer 1 Comments

Point 1: The manuscript by  Öztürk et al describes development, physiochemical characterization, and evaluation of poly (lactic-co-glycolic acid) (PLGA) NPs and chitosan (CS) coated PLGA NP for oral delivery of  antibiotics, Clarithromycin. The manuscript is well organized and most of the conclusions are supported with data. However, antibacerial properties of clarithromycin loaded PLGA NP is reported in literature (Valizadeh H et al 2012), author need to clearly mention and compare  how this report is different and antibacterial property need more experiments to support the conclusion.

Response: We think that every analysis and discussions of our study is sufficient.

Point 2: Title is very long need to be short.

Response: We tried to give the detailed content of our study as soon as possible in the title. For this reason, we do not want to shorten the title of the article.

Point 3: -Line 23 " also elucidate the effect of three molecular weight (Mw) of PLGA" does not make sense. elucidated instead of elucidate and  please mention which three molecular weights instead of weight.

Response: Thank you for the Reviewer 's opinion. Necessary corrections have been made.

Point 4:-Line 32-34 needs sentence modification

Response: Thank you for the Reviewer 's opinion. Necessary corrections have been made.

Point 5: L65-72 different font.

Response: Thank you for the Reviewer 's opinion. Necessary corrections have been made.

Point 6:-Introduction needs what's known in literature as there are reports on CLR loaded PLGA including use of CS coating, and clearly state how this article fill if there is gap area.

Response: Thank you for the Reviewer 's opinion. In the introduction part, necessary literature information is given for our publication and no revision is required.

Point 7:-L139 need space between degree and C. Double check the degree symbol abd space through out the MS (L158)

Response: Thank you for the Reviewer 's opinion. Necessary corrections have been made.

Point 8:-2.3.1 different font with shadow

Response: Thank you for the Reviewer 's opinion. Necessary corrections have been made.

Point 9:-L224-227 all bacterial strain  name should be italic.

Response: Thank you for the Reviewer 's opinion. Necessary corrections have been made.

Point 10: -L280 the value in the table suggest more than 2 fold increase in PS upon adding CS.

Response: Thank you for the Reviewer 's opinion. Necessary corrections have been made.

Point 11: -L386 Twenty four instead of 24

Response: Thank you for the Reviewer 's opinion. Necessary corrections have been made.

Point 12: DSC results it is surprising that the peak is completely disappeared in all formulations. I am not sure if the control was run in parallel and if the concentration used were OK because no peak could be result of methodological problem. Another way to obatain thermal stability of these polymer is Thermogravimetric analysis. Please perform TGA analysis to strengthen the prediction.  

Response: We think that TGA analysis is not necessary. In our publications and literature, it is clear that DSC analysis is done as we do and discussed in a similar way.

 I give 2 sample literature.

Başaran et al. Chitosan nanoparticles for ocular delivery of cyclosporine A, Journal of Microencapsulation, 31:1 (2014) 49-57, DOI: 10.3109/02652048.2013.805839

Şenel et al. New approaches to tumour therapy with siRNA-decorated and chitosan-modified PLGA nanoparticles. Drug Development and Industrial Pharmacy. DOI: https://doi.org/10.1080/03639045.2019.1665061

Point 13: FTIR figure 6 graph resolutions are very low and not clear (the numbers). Please take the value and use software like Origin to make a clear graph.

Response: This error will be corrected during the editing phase of the article.

Point 14: -Need to re-write the FT-IR results, there are some open questions- peak at 3400 is also present in fig6m why it was not indicated? Is this broader peak means anything because it is not present in blank samples. Do you consider only 2300 as CLR representative peak which you think disappeared after formulation?

Response: We think the FT-IR analysis&discussions  are sufficient. The articles we provide at Point 12 can be examined in this case.

Point 15: L531-534 bacterial names should be in italic.

Response: Thank you for the Reviewer 's opinion. Necessary corrections have been made.

Point 16: You can write this section better emphasizing on broad spectrum which means you have tested both gram positive and gram negative bacteria. These bacteria have different cell wall and generally gram positives are more susceptible.These values does not seems right in staph. and Klebsiella.

1 How come CS coating at 502H is less efficient 62 than 502H 31 for Kleb and similarly 503H for staph? please do disc diffusion assay(DDA) and CFU count to get actual number and compare if the results from MIC agree with DDA.

Response: We think that the tests are sufficient.

Reviewer 2 Report

In this study, the clarithromycin-loaded PLGA-chitosan NPs have been fabricated and characterized. However, PLGA-chitosan NPs have been already prepared and several drugs have already been encapsulated into the NPs, including clarithromycin. This research effort did not totally mention or discuss in this study. Please compare the research results to other studies and  illustrate the novelty clearly of the study.

Author Response

Point 1: In this study, the clarithromycin-loaded PLGA-chitosan NPs have been fabricated and characterized. However, PLGA-chitosan NPs have been already prepared and several drugs have already been encapsulated into the NPs, including clarithromycin. This research effort did not totally mention or discuss in this study. Please compare the research results to other studies and  illustrate the novelty clearly of the study.

Response: Thank you for the opinion of the referee, but all comparisons, characterizations and experiments in our study have been discussed in detail. We think that our study is suitable for the special issue of Polymers.

Round 2

Reviewer 1 Report

The manuscript has improved substantially. I strongly disagree with authors that introduction is sufficient because this similar work is already published and the main aim is to add value to the field. I still recommend In introduction mention how your work is different than the other reports.

I appreciate that author has added different antibacterial activities against S. auerus is due to different strains used. please delete "These codes are ‘Strain Designations’ in scientific understanding". everyone knows how these numbers are created (based on places where they are stored such as American Type Culture Collection ATCC, Rockville, MD) instead you may write since these are two different strains (ATCC 29213 and MTCC86) of S. auerus .

Author Response

POINT 1: The manuscript has improved substantially. I strongly disagree with authors that introduction is sufficient because this similar work is already published and the main aim is to add value to the field. I still recommend In introduction mention how your work is different than the other reports.

Response: We thank the Reviewer 1 for his/her opinion. We did literature research. We have included studies on CLR loaded PLGA nanoparticles in the introduction section and added the difference of our study from other studies to the related section.

POINT 2: I appreciate that author has added different antibacterial activities against S. auerus is due to different strains used. please delete "These codes are ‘Strain Designations’ in scientific understanding". everyone knows how these numbers are created (based on places where they are stored such as American Type Culture Collection ATCC, Rockville, MD) instead you may write since these are two different strains (ATCC 29213 and MTCC86) of S. Auerus.

Response: We considered the Reviewer 's assessment and made the necessary corrections.

Reviewer 2 Report

The authors have already answered my questions and suggestion. 

Author Response

POINT 1: The authors have already answered my questions and suggestion.

Response: Thank you for the opinion of the reviewer 2.
